# ITPKC as a Prognostic and Predictive Biomarker of Neoadjuvant Chemotherapy for Triple Negative Breast Cancer

**DOI:** 10.3390/cancers12102758

**Published:** 2020-09-25

**Authors:** Masanori Oshi, Stephanie Newman, Vijayashree Murthy, Yoshihisa Tokumaru, Li Yan, Ryusei Matsuyama, Itaru Endo, Kazuaki Takabe

**Affiliations:** 1Breast Surgery, Department of Surgical Oncology, Roswell Park Comprehensive Cancer Center, Buffalo, New York, NY 14263, USA; masa1101oshi@gmail.com (M.O.); snewman5@buffalo.edu (S.N.); Vijayashree.Murthy@RoswellPark.org (V.M.); Yoshihisa.Tokumaru@roswellpark.org (Y.T.); 2Department of Gastroenterological Surgery, Yokohama City University Graduate School of Medicine, Yokohama 236-0004, Japan; ryusei@yokohama-cu.ac.jp (R.M.); endoit@med.yokohama-cu.ac.jp (I.E.); 3Department of Surgery, Jacobs School of Medicine and Biomedical Sciences, State University of New York, Buffalo, New York, NY 14263, USA; 4Department of Surgical Oncology, Graduate School of Medicine, Gifu University, 1-1 Yanagido, Gifu 501-1194, Japan; 5Department of Biostatistics & Bioinformatics, Roswell Park Comprehensive Cancer Center, Buffalo, New York, NY 14263, USA; li.yan@roswellpark.org; 6Division of Digestive and General Surgery, Niigata University Graduate School of Medical and Dental Sciences, Niigata 951-8520, Japan; 7Department of Breast Surgery, Fukushima Medical University School of Medicine, Fukushima 960-1295, Japan; 8Department of Breast Surgery and Oncology, Tokyo Medical University, Tokyo 160-8402, Japan; 9Roswell Park Comprehensive Cancer Center, Elm & Carlton Streets, Buffalo, New York, NY 14263, USA

**Keywords:** biomarker, breast cancer, gene expression, metastasis, survival, ITPKC, TNBC, treatment response

## Abstract

**Simple Summary:**

Inositol 1,4,5-trisphosphate 3-kinase C (ITPKC) gene is a negative regulator of T cell activation and is a proven promoter of Kawasaki disease. Given the critical role of immune response in breast cancer, we aimed to determine the clinical relevance of ITPKC expression in breast cancer. ITPKC expression was highest in triple negative breast cancer, associated with its survival, and was its independent prognostic factor. Although high ITPKC expression was not associated with immune function nor with any immune cell fraction, low ITPKC triple negative breast cancer was significantly associated with activated cell proliferation, and they achieved a significantly rate in pathological complete response (disappearance of tumor) after neoadjuvant chemotherapy. To the best of our knowledge, this is the first report to demonstrate that ITPKC gene expression may be useful as a prognostic and predictive biomarker in triple negative breast cancer.

**Abstract:**

Triple negative breast cancer (TNBC) is the most aggressive subtype of breast cancer with higher mortality than the others. Pathological complete response (pCR) to neoadjuvant chemotherapy (NAC) is considered as a surrogate to predict survival. Inositol 1,4,5-trisphosphate 3-kinase C (*ITPKC*) is a negative regulator of T cell activation, and reduction in *ITPKC* function is known to promote Kawasaki disease. Given the role of tumor infiltrating lymphocytes in NAC and since TNBC has the most abundant immune cell infiltration in breast cancer, we hypothesized that the *ITPKC* expression level is associated with NAC response and prognosis in TNBC. The *ITPKC* gene was expressed in the mammary gland, but its expression was highest in breast cancer cells among other stromal cells in a bulk tumor. *ITPKC* expression was highest in TNBC, associated with its survival, and was its independent prognostic factor. Although high *ITPKC* was not associated with immune function nor with any immune cell fraction, low *ITPKC* significantly enriched cell proliferation-related gene sets in TNBC. TNBC with low *ITPKC* achieved a significantly higher pCR rate after NAC. To the best of our knowledge, this is the first report to demonstrate that *ITPKC* gene expression may be useful as a prognostic and predictive biomarker in TNBC.

## 1. Introduction

Breast cancer represents the most common malignancy and leading cause of cancer related morbidity and mortality all over the world, accounting for 30% of all new cancer diagnoses in females [1]. Triple negative breast cancer (TNBC) accounts for 10–15% of invasive cancers with an aggressive clinical course, early relapses and overall poor prognosis [2,3]. This is not only because TNBC lacks treatable molecular targets such as estrogen receptor (ER) or human epidermal growth factor receptor 2 (HER2) like the other subtypes [4], but also because TNBC is known to be highly proliferative [5] with a high expression of *Ki67*, the most commonly used marker of cell proliferation [6]. TNBC is known to relate to several proliferation-related pathways [7,8]. Our group has previously reported that TNBC has high activity of the E2F pathway [9] and G2M checkpoint pathway [10], both of which are essential components of the cell cycle and critical for cell proliferation.

Some TNBCs respond to neoadjuvant chemotherapy (NAC) (chemotherapy prior to definite operation), and pathological complete response (pCR) following NAC is now considered a surrogate to predict the long-term clinical outcome [11]. Given the aggressive clinical behavior and limited therapeutic options, a predictive biomarker for NAC response is needed not only to improve the efficacy of treatment, but also to reduce chemotherapy-related toxicity and improve quality of life.

Tumor infiltrating lymphocytes (TIL) have recently proven to play a major role in cancer biology and that includes breast cancer. One such example would be the significant role played by CD8+ effector T cells in the adaptive immune response through their production of interferon gamma (*IFN*- γ) and initiation of cytolytic activity [12]. This is especially true for TNBC which has an abundance of TIL [13]. Although it lacks clinical application, TIL has been reported as both a prognostic and predictive biomarker for pCR following NAC in TNBC, but not in hormone-positive breast cancer [14,15,16]. In *PD-L1*-positive metastatic TNBC, a pembrolizumab (anti-*PD-1* antibody) combined group demonstrated a significantly higher pCR rate than chemotherapy alone [17,18]. Atezolizumab (anti-*PD-L1* antibody) with Nab-pacritaxel was shown to have a significantly longer progression-free survival than Nab-paclitaxel alone in *PD-L1*-positive locally advanced and metastatic TNBC in the Impassion 130 study [19]. KEYNOTE-522 is a phase III study of NAC combined with pembrolizumab in patients with TNBC [20]. These reports suggest that, currently, *PD-L1* is used as a predictive biomarker for immune checkpoint inhibition, however, it is also known that there are many patients who respond to treatment without *PD-L1* expression. 

Our group has been reporting the clinical relevance of the tumor immune microenvironment (TIME) in breast cancer using bioinformatic algorisms with transcriptomic data, such as Gene Set Enrichment Analysis (GSEA) [21,22,23], CYBERSORT [24,25,26] and xCell [27,28,29].

Inositol 1,4,5-trisphosphate 3-kinase C (*ITPKC*), encoded by the *ITPKC* gene on chromosome 19q13.2, is known to be associated with Kawasaki disease, a pediatric disorder characterized by hyperactive T cells affecting the medium-sized arteries producing multi-systemic vasculitis predominantly seen in a Far East Asian population [30,31,32,33]. Recently, *ITPKC* expression was linked with cervical cancer carcinogenesis [34]. Overexpression of *ITPKC* was shown to increase microvascular adhesion of circulating colon cancer cells before hepatic metastasis formation [35]. Given that *ITPKC* is a negative regulator of T cells, we hypothesized that a low level of *ITPKC* expression is associated with enhanced cancer immunity, thus is associated with better response to NAC and survival in TNBC. 

## 2. Results

### 2.1. ITPKC Is Expressed in Mammary Gland, but Its Expression Was Highest in Breast Cancer Cells among Other Stromal Cells in a Bulk Breast Tumor

In order to identify which cells express *ITPKC* within a bulk breast cancer tumor, we analyzed *ITPKC* expression in tumor cells, B cells, stromal cells, myeloid cells and T cells using single-cell sequencing data of primary breast cancer (GSE75688) utilizing the same method we previously reported [36]. As shown in Figure 1, *ITPKC* was significantly expressed higher in breast cancer cells compared with stroma or immune cells. 

### 2.2. ITPKC Is Highly Expressed in TNBC and Is Associated with Its Worse Prognosis

We studied whether *ITPKC* expression is associated with clinical aggressiveness of breast cancer in The Cancer Genome Atlas (TCGA) cohort (Figure 2A). There was no significant difference in *ITPKC* expression by the American Joint Committee on Cancer (AJCC) cancer staging (Stage I–IV) and Nottingham grading system (Grade I, II and III) in a TCGA breast invasive carcinoma (BRCA) cohort (*p* = 0.681 and *p* = 0.824, respectively). *ITPKC* expression was significantly higher in TNBC compared to the other subtypes (ER-positive/HER2-negative, HER2-positive) (*p* = 0.027). 

In order to seek the clinical relevance of *ITPKC* expression, we performed survival analyses of the whole breast cancer cohort (Whole), and ER-positive/HER2-negative, TNBC and HER2-positive subtypes by Kaplan–Meier analyses of overall survival (OS), disease-free survival (DFS) and disease-specific survival (DSS) in the TCGA and DFS alone in the GSE25066 cohort (Figure 2B). Top quartile expression of *ITPKC* was defined as the high group, and the others as the low group. There was no association between any survival outcome and *ITPKC* expression in ER-positive/HER2-negative and HER2-positive subtypes in the TCGA, which were validated in the GSE25066 cohort. The HER2-positive subtype was not analyzed in the GSE25066 cohort because the cohort size was too small. High *ITPKC* expression was significantly associated with worse OS, DFS and DSS in TNBC (Figure 2B, all *p* < 0.001). This result was validated in the GSE25066 cohort that demonstrated a significant association of high *ITPKC* expression and worse DFS in TNBC (Figure 2B, DFS: *p* = 0.004 in TNBC). *ITPKC* expression was not associated with any survival outcome in the Whole cohort of TCGA, but high expression associated with worse DFS in the Whole cohort of GSE25066 (Figure 2B, DFS: *p* = 0.024). This is most likely a reflection of the disproportional ratio of TNBC in GSE25066 (178/467 = 38%) compared to the TCGA (159/1064 = 15%) or to the general population (10–15%). These findings suggest that high expression of *ITPKC* is significantly associated with worse survival in TNBC, but not with the other subtypes.

### 2.3. There Was no Difference in Clinical Characteristics between Low- and High-ITPKC Expression Groups in TNBC of TCGA

To study the association between *ITPKC* expression and clinical characteristics in TNBC, we compared *ITPKC* levels with age at diagnosis, race and tumor, using the tumor size, lymph node, and metastasis (TNM) as well as cancer stage (Stages I–IV) classification based on the AJCC cancer staging system in the TCGA (Table 1) as well as in the GSE25066 cohort (Appendix A). There was no statistically significant difference between the low- and high-*ITPKC* groups with any of the clinical characteristics analyzed in both cohorts.

### 2.4. ITPKC Expression Level Is an Independent Prognostic Factor for TNBC Survival

We then performed univariate and multivariate Cox regression analyses in TNBC of the TCGA cohort to investigate whether the prognostic value of the *ITPKC* expression level is independent of various clinical factors. We found that the *ITPKC* level is independent of clinical factors (such as age, race and TNM status). In the univariate analysis of disease-specific survival (DSS) in the TCGA cohort, AJCC cancer staging categories T, N and M, and *ITPKC* expression level had a significant hazard ratio (HR) (Table 2). In multivariate analyses, T, N and *ITPKC* were all found to be independent prognostic factors (HR = 2.50, 95% confidence interval (CI) = 1.17–5.34; *p* = 0.018).

### 2.5. ITPKC Expression Level Was not Associated with Immune-Related Pathway nor with Immune Cell Infiltration in TNBC

Since *ITPKC* is a negative regulator of T cell activity [32], we expected that *ITPKC* expression is associated with an immune-related pathway. To test this hypothesis, we performed the Gene Set Enrichment Analyses (GSEA) of Hallmark gene sets in both the TCGA and GSE25066 cohorts (Figure 3A). Surprisingly, *ITPKC* expression did not enrich any immune-related Hallmark gene sets: interferon (*IFN*)-α response, *IFN*-γ response, *IL2-STAT5* signaling and inflammatory response, in neither of the cohorts. 

Given the function of *ITPKC* to suppress T cells [32,37], it was of interest to investigate the association between *ITPKC* expression and immune cells composition utilizing the xCell algorithm. Again, we found that high-*ITPKC* expression tumors were not associated with any T cells fractions, including CD8 T cell, CD4 memory T cell, T helper 1 (Th1) and regulatory T cell, except for T helper 2 (Th2) cells, in the TCGA cohort (Figure 3B). A high-*ITPKC* tumor was significantly associated with a high fraction of CD8, regulatory T and a low fraction of Th1 in the GSE25066 cohort. These results combined, there was no immune cell fraction that was consistently associated with *ITPKC* expression levels in both cohorts, thus we concluded that there is no association between *ITPKC* expression and immune cell infiltration in TNBC. 

### 2.6. Low-ITPKC Expression Tumors Enriched Cell Proliferation-Related Gene Sets in TNBC

To investigate why low-*ITPKC* expression TNBC is associated with better survival, we performed GSEA of the 50 Hallmark gene sets to low-ITPKC expression tumors. Low-*ITPKC* TNBC significantly enriched three out of five Hallmark cell proliferation-related gene sets: G2M checkpoint, E2F targets and mitotic signaling (Figure 4A; normalized enrichment score (NES) = −1.83, −1.69, −1.92, and FDR = 0.09, 0.15, 0.11, respectively). These results were validated with the GSE25066 cohort (Figure 4A; NES = −1.46, −1.49, −1.58, and FDR = 0.16, 0.16, 0.12, respectively). Low expression of *ITPKC* was significantly associated with high expression of *MKI67*, the most commonly used marker of cell proliferation in the clinical setting, in the TCGA cohort (Figure 4B, *p* < 0.001). However, this was not validated with the GSE25066 cohort (*p* = 0.806). The proliferation score derived from a previous report [38] was significantly elevated in low-*ITPKC* expression TNBC (*p* < 0.001) (Figure 4C).

### 2.7. Pathological Complete Response (pCR) was Associate with Lower Expression of ITPKC and Low ITPKC Is Predictive of pCR to Neoadjuvant Chemotherapy (NAC) in TNBC

It is well known that highly proliferative cancers are more likely to respond to cytotoxic chemotherapy [9,10,39,40]. Given our results that low-*ITPKC* expression TNBCs are highly proliferative, we hypothesized that they are associated with better NAC response. We found that patients who achieved pCR after NAC had significantly lower *ITPKC* expression in TNBC, but not in the ER-positive/Her2-negative subtype, consistently in two cohorts, GSE25066 and GSE20194 (Figure 5A; TNBC *p* = 0.015, 0.004; ER-positive/Her2-negative *p* = 0.118, 0.395, respectively). 

We further found that TNBCs with low *ITPKC* expressions prior to NAC were significantly associated with an increased likelihood of achieving pCR compared to those with high *ITPKC* expression in GSE25066 (*p* = 0.005), which was validated in GSE20194 (*p* = 0.021) (Figure 5B). Only one cohort (GSE25066) revealed a similar association of low *ITPKC* expression and high rate of pCR in ER-positive/HER2-negative tumors (*p* = 0.002); however, this result was not validated in GSE20194 (*p* = 0.189). The above findings suggest that expression of *ITPKC* has a predictive role in neoadjuvant treatment response in TNBC.

## 3. Discussion

The *ITPKC* gene was highly expressed in the breast mammary gland, but its expression was found to be highest in breast cancer cells among other stromal cells in a bulk tumor. TNBC expressed the highest levels of *ITPKC* among subtypes. High *ITPKC* expression was significantly associated with worse survival in TNBC but not in the other subtypes. *ITPKC* expression was found to be an independent prognostic factor in TNBC. Although high *ITPKC* was not associated with immune function nor with any immune cell fraction, low *ITPKC* significantly enriched cell proliferation-related gene sets in TNBC. A low-*ITPKC* tumor was significantly associated with a high pCR rate in TNBC. To the best of our knowledge, this is the first report to demonstrate that *ITPKC* expression may be useful as both a prognostic and predictive biomarker in TNBC.

*ITPKC* is one of the three isoenzymes of inositol 1,4,5-trisphosphate 3-kinase (*ITPK*) that phosphorylate inositol 1,4,5-trisphosphate (*IP3*), a key second messenger in many cell types [41]. *ITPKC* regulates immune responses [32], such as modulation of nuclear factor of activated T cells [42,43]. The biological impact of *ITPKC* on disease pathogenesis has been most widely studied in Kawasaki disease, Hirschsprung disease and more recently in cervical squamous cell carcinoma and colorectal malignancies with hepatic metastases [32,34,35,44]. In a study of 465 patients with cervical squamous cell carcinoma and 800 controls, Yang et al. found that the G/G genotype and G allele of the *ITPKC* rs28493229 polymorphism contributed to carcinogenesis of cervical cancer, whereas the C/G genotype and C allele protected against disease development [34]. Using random homozygous gene perturbation (RHGP) in combination with specific adhesion assays of cancer cells, expression changes of *ITPKC* were found to regulate hepatic microvascular adhesion of circulating colon cancer cells by overexpression in nonadherent cancer cell clones [35]. We were unable to find any study other than ours to report the association of *ITPKC* expression with cell proliferation in breast cancer. 

A predictive biomarker to select responders to NAC is expected to maximize the treatment benefit, reduce toxicities and financial costs and improve patients’ quality of life. Since NAC is known to be effective in highly proliferative cancer, we have previously focused on the generation of scores that reflect cell proliferation as biomarkers, that predict NAC response to breast cancer. We found that the E2F pathway plays a critical role in the cell cycle predicted NAC response in ER-positive/HER2-negative but not in TNBC patients [9]. We further found that the G2M checkpoint pathway is also critical in the cell cycle predicted NAC response in ER-positive/HER2-negative but not in TNBC patients [10]. We have established a four-gene score from the genes differentially expressed between the parental and lung metastasis cell lines that predicted NAC response in ER-positive/HER2-negative, and again, not in TNBC patients. In the current study, we found that ITPKC expression predicts the NAC response in TNBC with a single gene expression, which is far more clinically appreciable in terms of cost and simplicity. Genomic signature profiling, such as Oncotype Dx and MammaPrint, has been used in the clinical practice to predict the benefit of adjuvant chemotherapy in hormone-positive breast cancers. Since *ITPKC* expression predicts the NAC response of TNBC, it does not overlap with the existing genomic signature profiling. We cannot help but speculate that the *ITPKC* expression may have a clinical utility to be used for patient selection and as a predictive biomarker for NAC in TNBC. 

Although we found a novel role of *ITPKC* expression in TNBC, this study is not free from limitations. First, our analysis is a retrospective study and limited in the measurement of gene expression. Secondly, our study was conducted by a bioinformatics approach alone. Although we have validated our findings using multiple cohorts in silico, in vivo and in vitro experimental approaches are needed to elucidate the mechanism as well as the causality of low expression of *ITPKC* and cell proliferation. Further, since we use publicly available multiple cohorts that are often from completely different sizes and backgrounds, the results may vary. Finally, in order to use *ITPKC* expression in TNBC management as both a predictive and prognostic biomarker, we need to analyze using quantitative polymerase. 

In conclusion, we demonstrated that low *ITPKC* expression in TNBC was significantly associated with better survival and better response to NAC, which was not the case in the ER-positive/HER2-negative subtype. Thus, *ITPKC* expression may be useful as a prognostic and predictive biomarker in TNBC.

## 4. Materials and Methods

### 4.1. Cohorts Used for Analyses

Boxplots were generated using R version 4.0.1. Tumor RNA sequencing-based expression and clinical data for 1065 patients of The Cancer Genome Atlas (TCGA) breast cancer (BRCA) project [45] were obtained from the cBio Cancer Genomic Portal [46]. The TCGA was chosen because it is one of the largest cohorts with a full transcriptome that is attached to robust clinical information. Single-cell sequencing data of primary breast cancer were obtained from the studies of Chung et al. (GSE75688) [47]. Clinicopathologic and normalized microarray-based gene expression data were also obtained for the studies of Hatzis et al. (GSE25066; *n* = 467) [48], Shi et al. (GSE20194; *n* = 248) [49] and Noguchi et al. (GSE32646; *n* = 81) [50] from the GEO repository (http://www.ncbi.nlm.nih.gov/geo). These cohorts were chosen because they have clinical response data after neoadjuvant chemotherapy that TCGA lacks. Gene expression levels in the TCGA, such as *ITPKC*, were directly obtained from the cBio Portal, and the average of the gene expression levels was used when there were multiple probes for a single gene in the gene expression microarray, which was transformed to log2 for analyses as we previously described [36,51]. In terms of survival data, overall survival (OS), disease-free survival (DFS) and disease-specific survival (DSS) were available in the TCGA, and DFS was available in the GSE25066 cohort. 

### 4.2. Cell Composition Fraction Estimation

xCell, a computational algorithm, was used to estimate the cell score of a tumor from its gene expression profiles [52]. Calculated data were downloaded through the xCell website (https://xcell.ucsf.edu/).

### 4.3. Gene Set Expression Analyses

Gene Set Enrichment Analyses (GSEA) software (Java version 4.0) [53] with MSigDb Hallmark was used [54] for gene set enrichment analysis, and a false discovery rate (FDR) of 0.25, as recommended by the GSEA software, was used to deem statistical significance as we previously reported [23,27,28,29,36,55,56,57,58,59,60,61].

### 4.4. Other

Statistical analyses and data plotting were performed using R (version 4.0.1) or Microsoft Excel (version 16 for Windows). The value of 0.05 was the *p* value cut-off for statistical significance. The Kaplan–Meier method with a log rank test was used for survival analysis. Two-tailed Fisher’s exact tests or one-way ANOVA provided statistical comparisons between groups. Tukey-type boxplots demonstrate median and inter-quartile level values. 

## 5. Conclusions

Although we did not see any correlation with immune cell number or activity, we found that low *ITPKC* gene expression was associated with highly proliferative activity in TNBC and hence better response to neoadjuvant chemotherapy.

## Figures and Tables

**Figure 1 cancers-12-02758-f001:**
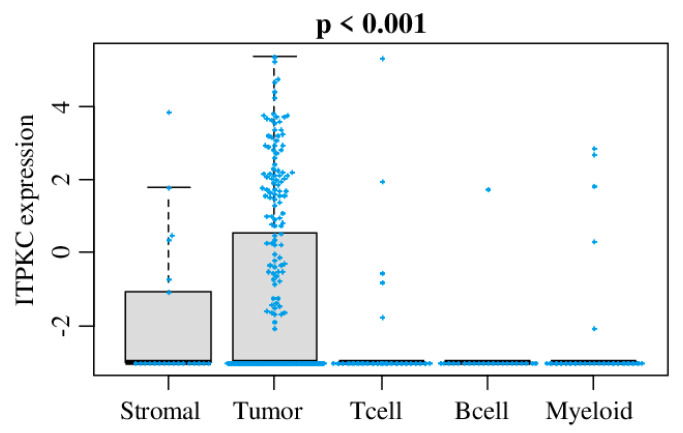
*ITPKC* gene expression among cells in the tumor microenvironment. Boxplots of the *ITPKC* gene expression by cancer cells, stromal cells, T cells, B cells and myeloid cells in single-cell sequencing data of primary breast cancer in GSE75688. One-way ANOVA test was used to calculate *p* values. Tukey-type boxplots show median and inter-quartile level values.

**Figure 2 cancers-12-02758-f002:**
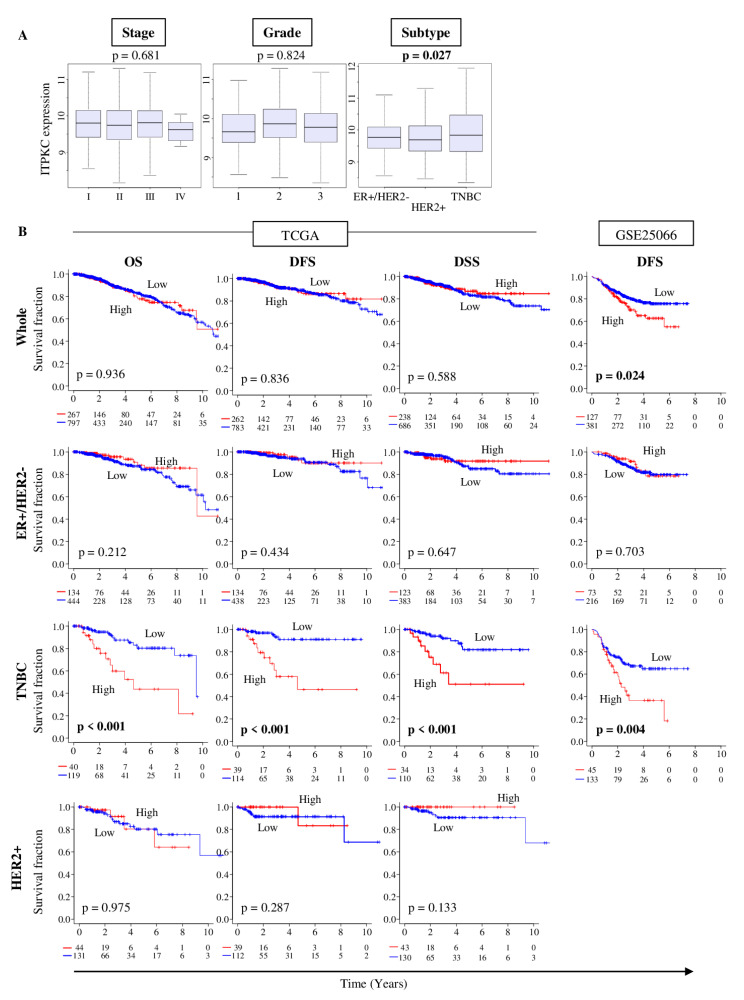
*ITPKC* gene expressions and clinical features including survival in the TCGA and GSE5066 breast cancer cohort. (**A**) Boxplots of the ITPKC expression by American Joint Committee on Cancer (AJCC) stage, Nottingham pathological grade and breast cancer subtype. One-way ANOVA test was used to calculate *p* values. Tukey-type boxplots show median and inter-quartile level values. (**B**) Association between *ITPKC* expression and survival of breast cancer patients in the TCGA and GSE25066 cohorts. Overall survival (OS), disease-free survival (DFS) and disease-specific survival (DSS) of *ITPKC* expression low and high in whole breast cancer cohort (Whole), estrogen receptor-positive/human epidermal growth factor receptor 2-negative (ER+/HER2-), triple negative breast cancer (TNBC) and HER2+ in the TCGA cohort and DFS in the GSE25066 cohort. The top quartile was defined as the high-score group within each cohort. Log rank test was used to compare between two groups with Kaplan–Meier survival curves and to calculate *p* values.

**Figure 3 cancers-12-02758-f003:**
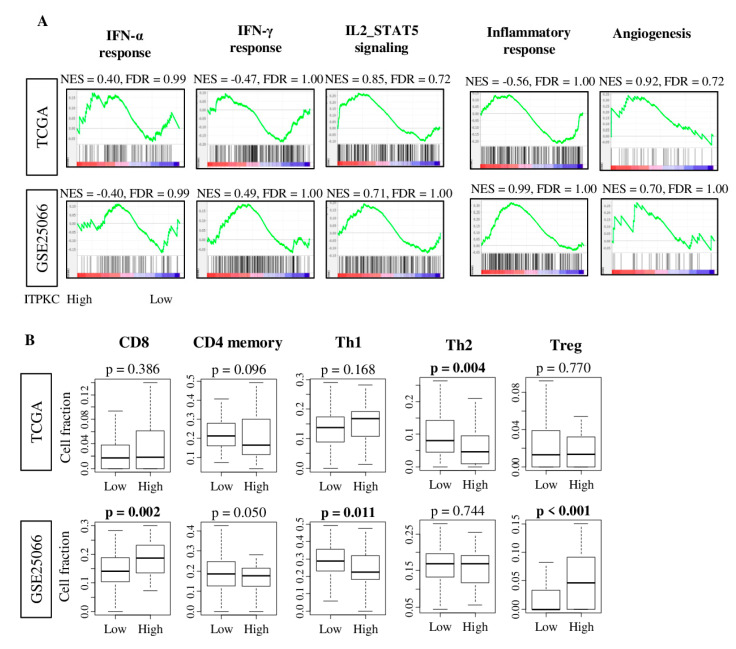
*ITPKC* gene expression and immune function and immune cell fraction with TNBC in the TCGA and GSE25066 cohorts. (**A**) Correlation plots of immune-related gene sets: interferon (*IFN*)-α and γ, *IL2-STAT5* signaling, *IL6/JAK/*STAT3 signaling, inflammatory response and angiogenesis, with normalized enrichment score (NES) and false discovery rate (FDR). FDR < 0.25 is considered to be statistically significant. (**B**) Boxplots of the comparison with CD8, CD4 memory, T helper type 1 cells (Th1), T helper type 2 cells (Th2) and regulatory T cell (Treg), by low and high *ITPKC* expression in both cohorts. The top quartile was defined as the high-score group within each TNBC cohort. One-way ANOVA test was used to calculate *p* values. Tukey-type boxplots show median and inter-quartile level values.

**Figure 4 cancers-12-02758-f004:**
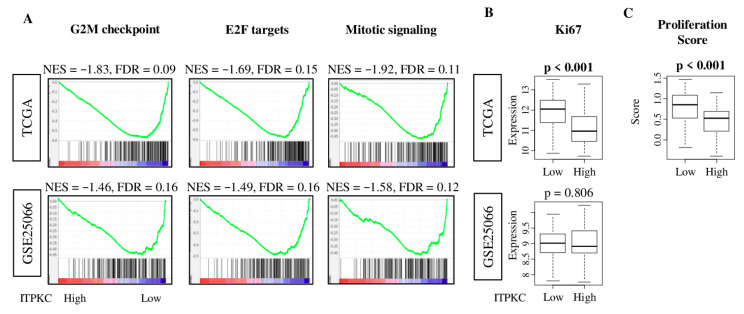
Hallmark gene sets with significant enrichment in low-*ITPKC* group of TNBC in both TCGA and GSE25066 cohorts. (**A**) Gene set enrichment plots along with NES and FDR of gene sets: G2M checkpoint, E2F targets and mitotic signaling. NES and FDR were determined with the classical GSEA method. (**B**) *MKI67* gene expression in the low- and high-*ITPKC* expression groups of both cohorts. (**C**) Boxplots of proliferation score between low and high *ITPKC* expression in the TCGA cohort. The top quartile was defined as the high-score group within each cohort. One-way ANOVA test was used to calculate *p* values. Tukey-type boxplots show median and inter-quartile level values.

**Figure 5 cancers-12-02758-f005:**
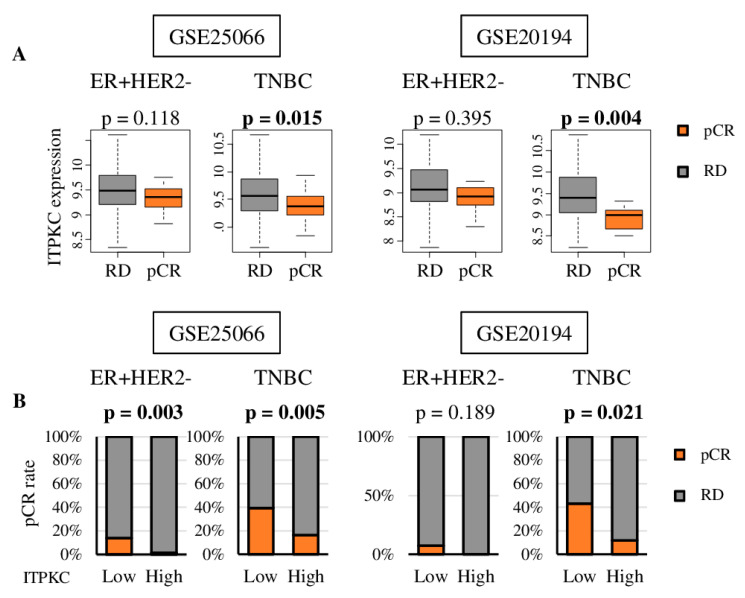
*ITPKC* gene expression and treatment response in ER+/HER2- and TNBC patients. (**A**) Boxplots of the *ITPKC* expression of residual disease (RD) and pathological complete response (pCR) after NAC in GSE25066 (*n* = 467) and GSE20194 (*n* = 197) cohorts. One-way ANOVA test was used to calculate *p* values. (**B**) pCR rate after NAC by low and high *ITPKC* expression in ER+/HER2- and TNBC in same two cohorts. The top quartile was defined as the high-score group within each cohort. Fisher’s exact test was used to calculate *p* values.

**Table 1 cancers-12-02758-t001:** Clinical characteristics of low and high ITPKC expression in TNBC patients in the TCGA cohort.

Clinical Variables	*ITPKC*-Low	*ITPKC*-High	*p*-Value
	(*n* = 119)	(*n* = 40)	
Age at diagnosis			0.920
Median	55.0	52.5	
IQR	48–62	44–66	
Race			0.078
White	75 (63.0%)	15 (37.5%)	
Black	29 (24.4%)	25 (62.5%)	
Asian	8 (6.7%)	0 (0%)	
AJCC T-category			0.116
T1	28 (23.5%)	12 (30.0%)	
T2	79 (66.4%)	20 (50.0%)	
T3	10 (8.4%)	5 (12.5%)	
T4	2 (1.7%)	3 (7.5%)	
AJCC N-category			0.440
N-	81 (68.1%)	24 (60.0%)	
N+	38 (31.9%)	16 (40.0%)	
AJCC M-category			0.405
M-	104 (87.4%)	30 (75.0%)	
M+	1 (0.8%)	1 (2.5%)	
Stage at diagnosis			0.078
I	20 (16.8%)	8 (20.0%)	
II	82 (68.9%)	20 (50.0%)	
III	14 (11.8%)	10 (25.0%)	
IV	1 (0.8%)	1 (2.5%)	

AJCC: American Joint Committee on Cancer, IQR: interquartile range.

**Table 2 cancers-12-02758-t002:** Survival analyses of ITPKC expression and other factors in the TCGA cohort.

TCGA (DSS)	Univariate		Multivariate	
	HR (95% CI)	*p*-Value	HR (95% CI)	*p*-Value
Age	1.43 (0.46–4.40)	0.536		
Race (Caucasian vs. other)	0.51 (0.19–1.35)	0.177		
T (T3/4 vs. T1/2)	7.16 (2.62–19.59)	<0.001 *	2.46 (0.59–10.19)	<0.001 *
N (N+ vs. N-)	5.36 (1.89–15.23)	0.001 *	15.42 (3.72–63.86)	0.002 *
M (M+ vs. M-)	9.43 (2.13–41.71)	0.003 *	0.61 (0.08–4.57)	0.631
*ITPKC* expression level	1.97 (1.03–3.76)	0.041 *	2.50 (1.17–5.34)	0.018 *

CI: confidence interval, DSS: disease-specific survival, HR: hazard ratio. * *p*-value < 0.05.

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
