# Peer review of "ITPKC as a Prognostic and Predictive Biomarker of Neoadjuvant Chemotherapy for Triple Negative Breast Cancer"

_cancers, 2020, doi:10.3390/cancers12102758_

Round 1

Reviewer 1 Report

The manuscript results well designed, descripted and organized.

Since relevant prognostic and predictive biomarkers in TNBC patients are still missing, any new contribution, mostly in neoadiuvant setting, could be relevant, already with the related methodological limitations of this work, as the authors referred.

Some minor considerations to improve the manuscript have been reported as follow:

  1. In lines 64-66, the authors indicated 17 self-citations. It looks excessive compared the total number of references reported. The most relevant should be chosen.
  2. In lines 57 the authors mentioned the Tumor infiltrating lymphocytes (TIL) as relevant biomarker in breast cancer. Some mentions regarding the role of PDL-1 biomarker and the recent improvements in term of response to checkpoint inhibitors in TNBC should be reported in the introduction (Schmid et al. 2018, Schmid et al. 2020, Rugo et al. 2019, etc.)
  3. In lines 81-82, 209-210, 217 there are mistakes in the caracters used
  4. In fig. 5B there is a mistake, pCR rate and RT have to be reported in Fig. 5A
  5. In methods, the references 51 and 53 do not match with those mentioned in the text
  6. Reference should be references

Reviewer 2 Report

The team has a extensive research work and many recent publications on similar topics in high impact journals.

The topic is highly actual and of interest not only from scientific research point of view but from impact on patients' outcome and identification of those patients who may benefit most from NAC.

Research methodology is rigorous and extensively described. It is also connected to other publications of the research group, that are cited in the References section.

Conclusion are concise and of interest and impact for future research and, perhaps, clinical practice.

However I consider that some minor corrections are necessary.

I suggest that section Material and Methods must be placed after Introduction section and not after Discussion. It is thus much easier for the reader to understand the Results section.

Perhaps that a table with some of the demographic data of the study cohorts of interest for the study would be of help to understand the differences found in Results between cohorts; something similar for Table 1 and 2 that include clinical data of TCGA only.

Introduction

Rows 70-71. The sentence "A functional single nucleotide...." should be deleted since this is not necessarily related to the topic of the study.

Results

Page 2 rows 80-86. I suggest deleting this paragraph along with Figures 1A and B since this is not the topic of the article. Eventually it may be restrain to a phrase. Manuscript has enough data to process even without this paragraph.

Material and methods. I suggest a phrase on why (reasons) these cohorts have been included in this analysis. How were these cohorts chosen? What was the rationale behind choosing these cohorts?

A short description of the methods of investigation (e.g. ITPKC) must be provided even if the studies are cited. This would be of help for an easier follow-up of the method and results.

Page 12.row 286. Chung's study is not cited. Please add citation.

References

References should be drafted in the same style. References 21-25 (for example) include capital letters in the title.

Discussion

Discussion on the potential reasons for differences in results between cohorts would be of help to understand better the results.
